# Efficient AI-driven allegation screening: A case study of Thailand's National Anti-Corruption Commission

Issara Sereewatthanawut[1,2]*, Patipan Sriphon[1], Pattrawut Khunwipusit[1], Babatunde Oluwaseun Ajayi[1], Ademola Enitan Ilesanmi[3], Jutarat Suwaree[4], Wonlop Writthym Buachoom[5]

1 King Prajadhipok's Institute, Bangkok, Thailand, 2 Bangkokthonburi University, Bangkok, Thailand, 3 University of Pennsylvania, Philadelphia, United States of America, 4 Office of the National Anti-Corruption Commission, Thailand, 5 Business School, King Mongkut's Institute of Technology, Ladkrabang, Thailand

* issara.se@kpi.ac.th

## Abstract

Efficient screening of corruption allegations is crucial for promoting accountability and transparency in public administration. However, many institutions still rely on manual processes that are prone to inefficiency and inconsistency. As AI gains traction across sectors, this study develops and evaluates an artificial intelligence (AI)-powered prototype designed to support the preliminary screening of corruption complaints at Thailand's National Anti-Corruption Commission (NACC). The proposed system integrates Optical Character Recognition (OCR), Natural Language Processing (NLP), and machine learning techniques to automate document handling and improve workflows. A mixed-methods research approach was adopted, combining institutional process analysis with a comprehensive technical performance assessment. The OCR module achieved an F1-score of 81.8%, with precision and recall of 84.2% and 79.6%, respectively. For printed text, the system attained 72% word-level accuracy and 78% at the character level. Additionally, the integrated framework demonstrated a classification accuracy of 57.5% and significantly improved operational efficiency, reducing average complaint processing time by 78.6% compared to traditional manual methods. The findings highlight AI's transformative potential in enhancing anti-corruption efforts through increased speed, accuracy, and consistency. They underscore the importance of responsible and context-sensitive AI adoption in public sector governance. This study contributes to the growing discourse on digital governance by providing empirical evidence and practical insights for policymakers and practitioners aiming to implement scalable, transparent, and ethically grounded AI solutions within institutional accountability frameworks.

**Data availability statement:** The codes and related sources for this project can be accessed at https://github.com/VAP-Solution/kpi_accusation_fieldtest, which contains materials related to the field test. The test employed a custom web application developed using Appsmith, available at https://www.vapsolution.app/nacc_llm_fieldtest, which served as the primary interface for testing and data collection. However, the research data cannot be made publicly available due to data protection agreements with the data provider, the National Anti-Corruption Commission (NACC) of Thailand.

**Funding:** National Research Council of Thailand.

**Competing interests:** The authors have declared that no competing interests exist.

## Introduction

Corruption, generally defined as the use of public office for personal gain, presents a significant challenge to good governance, equitable economic growth, and social justice [1]. Its widespread impact is recognized as a global problem, with serious implications for institutional integrity and the provision of public services. Consequently, corruption continues to draw considerable scholarly attention [2,3]. Effectively combating corruption is not just an ethical concern. It is essential for building institutional resilience, supporting sustainable growth, and upholding the rule of law [4]. In this regard, anti-corruption agencies play a vital role.

In Thailand, the National Anti-Corruption Commission (NACC) leads this effort, responsible for receiving, investigating, and judging allegations of corruption [5]. Despite its strategic goal, the NACC's effectiveness is limited by outdated procedures. Its complaint-handling system is largely manual, causing systemic inefficiencies. These include long case processing times, higher administrative costs, and inconsistencies in initial assessments. Internal audits show that the average time to resolve a complaint is about 73.5 days, more than twice the organization's standard of 30 days [5]. Such delays erode public trust and weaken the commission's ability to respond promptly to corruption issues.

Persistent inefficiencies within public institutions have led to increasing interest in the application of artificial intelligence (AI) to enhance administrative effectiveness and procedural integrity [6,7]. Recent studies highlight the capacity of AI in governance. The study of [8] emphasizes the integration of machine learning (ML) and AI technologies into public policy processes, ensuring transparency and clarity. Similarly, the studies [9,10] highlight the essential role of AI in improving transparency and accountability in public governance systems. Globally, public integrity systems are increasingly leveraging AI technologies to detect fraudulent patterns and anomalies in large datasets [11]. These trends show considerable promise for institutions like Thailand's NACC to benefit from AI implementation, especially in optimizing case screening and management through intelligent automation. AI can automate routine and resource-intensive tasks, such as digitizing documents, extracting content, and categorizing cases. Reducing manual processes can help prevent delays, minimize inconsistencies due to human subjectivity, and improve the accuracy and speed of initial assessments. This transition is not only possible but also fits well with Thailand's national digital governance agenda, which encourages transparency, accountability, and data-driven decisions in the public sector [12]. Despite growing research on AI-powered transparency and accountability in governance, limited studies have operationalized and empirically evaluated AI systems within anti-corruption institutions. This research aims to fill this gap by developing and evaluating an AI-driven system designed to improve the efficiency and transparency of corruption complaint screening at Thailand's NACC. However, integrating AI into existing legal and regulatory frameworks is complex and requires careful attention. Main challenges include protecting data privacy, ensuring algorithms are transparent, making systems scalable, and training staff to work with AI tools. In addition, the lack of specific legal provisions for AI-assisted decision-making creates apprehension regarding legitimacy and sustainability [11].

To address this issue, this study designs, implements, and evaluates an AI-driven prototype aimed at improving the preliminary screening of corruption allegations submitted to the NACC. The system integrates Optical Character Recognition (OCR), Natural Language Processing (NLP), and Machine Learning (ML)–based classification to automate key stages of the complaint intake process. It employs general-purpose large language models (LLMs), Gemini-1.5-Flash and Gemma27B, for text classification through prompt-based inference without additional training or fine-tuning. Task-specific prompts guide classification decisions, enabling efficient and consistent case categorization.

A mixed-methods research design was adopted, combining institutional analysis with a technical performance evaluation to assess improvements in processing speed, accuracy, and user satisfaction. From these perspectives, this study contributes to the growing body of knowledge on AI adoption in public sector governance and anti-corruption systems. It also provides actionable insights for policymakers and practitioners on how to deploy intelligent systems responsibly and effectively to strengthen institutional integrity and operational efficiency.

## Literature review

Corruption remains a major hindrance to the realization of good governance, particularly in developing nations' institutional arrangements [13]. In Thailand, the NACC is mandated by law to receive, investigate, and act on complaints of corrupt practices [13,14]. However, despite this statutory provision, the complaint handling system of the Commission is still largely manual. While multiple channels may be utilized for public submission—e.g., hardcopy forms, email correspondence, and the online Public E-Service for Complaint and Allegation (PESCA) platform—the subsequent procedures of preliminary screening, classification, and triage still remain largely undertaken by human officers. This dependence on human intervention results in several procedural inefficiencies, such as long screening durations (with an average processing time of 73.5 days), unequal workload distribution across officers, and lack of consistency in interpretive judgments. Collectively, these shortcomings undermine procedural justice and erode public confidence in the responsiveness and institutional integrity of the anti-corruption system [15,16].

The preliminary screening of corruption complaints within the existing scheme is a series of stages, including receipt of documents, reading content, jurisdictional analysis, identification of allegation type, and subsequent internal routing. These processes involve manual extraction of factual elements and subjective assessment of completeness as well as legal merit of each submission. This process is unavoidably time and labor-intensive and thus is susceptible to inconsistency and procedural delay [17]. The widespread reliance on individual discretion not only introduces the risks of interpretive bias but also contributes to systemic delay and the accumulation of undecided cases. While the Public E-Service for Complaint and Allegation (PESCA) platform has ushered in efficiencies in logistical intake and case registration, it lacks embedded intelligence to support automated classification, prioritization, or decision support [17]. The screening process, therefore, remains mired in inefficiencies that compromise the operational agility as well as procedural fairness of the anti-corruption apparatus.

Artificial Intelligence (AI) has increasingly become a key driver in the modernization of public administration, with spectacular promises of enhancing the efficiency, transparency, and responsiveness of government operations. Within the range of AI-driven technologies, Optical Character Recognition (OCR), Natural Language Processing (NLP), and Machine Learning (ML) have shown tremendous promise in the automation of routine bureaucratic procedures in strategic sectors such as tax auditing, healthcare, and legal documentation [18,19]. In the specific scenario of public complaint management, these technologies are being strategically leveraged to minimize the challenges involved in labor-intensive and time-consuming administrative processes, including data entry, document classification, and case prioritization [20]. OCR plays a critical role in facilitating the digitization of unstructured information, such as handwritten submissions and scanned documents, typically relating to complaints filed by citizens. Intelligent triage is significantly supported when coupled with NLP approaches that have the ability to recognize named entities and discern contextual meanings. This includes summarization of key issues and as well as the automatic routing of grievances to appropriate legal or

administrative departments [21]. In addition, recent developments in transformer-based NLP models have enhanced the capacity of such systems to comprehend and process the complex, often esoteric language characteristic of bureaucratic and legal correspondence. Such developments position AI as an effective and powerful tool to navigate the procedural intricacies involved in administrative governance [22].

In recent times, the use of Artificial Intelligence (AI) in supporting decision-making in the public sector has gained attention, with broad applications in legal reasoning, fraud identification, and administration monitoring. For instance, research by [23] applied machine learning algorithms to learn from Indonesian procurement information and effectively identify irregular bidding patterns indicative of potential collusion or corruption. informing this empirical endeavor, the research of [24,25] examined the normative dimensions of AI adoption, bringing into focus the ethical concerns, as well as the governance requirements necessary for its effective and secure application in legal and administrative domains. These efforts illuminate the revolutionary potential and the inherent limitations of AI in public administration, while underscoring the urgency of making transparency, fairness, and explainability a reality, particularly in high-stakes areas like corruption screening.

The theoretical basis of the present study is embedded in public value theory, which argues that technological innovation in the public sector must go beyond efficiency or effectiveness gains and address how they strengthen institutional legitimacy and support democratic norms [26]. Despite policy efforts to support digital transformation, as exemplified by local plans like Thailand's e-Government Master Plan and strategic AI roadmap developed through the Digital Government Development Agency's (DGA), actual adoption of AI in the frontline of public complaint processing continues to be limited [16]. Current use is largely driven towards investigative analytics and post-hoc processing of information, with a considerable gap remaining in the "first mile" of anti-corruption processes. This study seeks to address that gap through the development of an AI-supported system, particularly designed to assist in the intake, classification, and forwarding of corruption reports to promote a more responsive and accountable governance.

Therefore, this study proposes a modular AI-based system to address the text-dominant and classification-oriented needs of public grievance screening. The designed system incorporates three integrated modules, each of which is tailored to a specific step of the institutional process of the NACC. An OCR module enabling scanned documents and handwriting inputs, facilitating machine-readable text capture, based on a locally adapted Tesseract engine version [27], is enabling the digitization of unstructured inputs prevalent in citizen complaints as an initial step. The NLP module then leverages advanced transformer-based architectures to perform a range of linguistic processing tasks, including named entity recognition, syntactic parsing, and semantic summarization of complaint narratives [28,29]. This layer enables the extraction of legally relevant information and the reduction of textual complexity, thereby facilitating downstream decision-making tasks. Third, the classification module employs an ML-based system to automate key aspects of the complaint intake process. The classification task is framed as a multi-class prediction problem given as $\hat{y} = argmax_{y_i}P(y_i|x)$, where $x$ is the feature vector extracted from the complaint message after NLP processing, and $y_i$ is the set of pre-defined allegation categories, e.g., bribery, abuse of authority, or conflict of interest. The model predicts the highest conditional probability class label $\hat{y}$, thereby facilitating computerized routing of every complaint to the relevant investigating or legal unit. Such an end-to-end design not only reflects the procedural order of the NACC's complaint handling mechanism but also introduces intelligence at every stage in the workflow, leading to a scalable and open platform for institutional responsiveness and procedural justice.

This study identifies three dimensions of research that collectively tackle the technical efficiency and institutional implications of AI application in processing public grievances against corruption. First, it examines the extent to which the application of AI technologies can reduce the mean processing time in the initial assessment of corruption-related complaints filed with NACC. This study aims to measure the efficiency gains brought about by the automation of a traditionally human-intensive bureaucratic process. Next, the study evaluates the accuracy of classification performance brought about by the proposed AI model, with specific focus on its ability to consistently identify and classify various types of corruption

allegations from unstructured text data. The study focuses on assessing the system's precision, F1 score, recall, and overall accuracy in replicating expert-level decision-making results within a multi-class classification scheme. Finally, the analysis extends beyond algorithmic performance to consider the broader operational and institutional implications of AI adoption. In particular, it considers possible improvements in workload distribution, procedural consistency, and administrative efficiency, thereby situating the technological intervention within the normative expectations of transparency, fairness, and public sector responsiveness. As such, these research questions are designed to impart a holistic appreciation of AI's transformative potential within the initial adjudication processes of anti-corruption administration. To enhance clarity, the main points from the literature review are summarized in Table 1.

## Methodology

In response to contextual applicability and methodological dependability, the present study utilized a mixed-methods design that combined qualitative institutional study with the development and empirical verification of an AI system. The research is conducted in three successive stages: (1) institutional process analysis and description of the problem, (2) developing a prototype system, and (3) system testing and performance assessment.

### The institutional process

The initial phase of the study was dedicated to in-depth analysis of the procedural and structural designs of the existing complaint-handling system of the NACC. The initial phase utilized qualitative methods, including in-depth interviews and focus group discussions with 25 strategically selected stakeholders, including NACC officials, ICT experts, and lawyers. These interactions enabled structured mapping of institutional procedures and the identification of main inefficiencies in the form of procedural delays, uncertainty in jurisdictional authority, and variation in document handling practices. These interactions also provided nuanced insights into institutional expectations of incorporating digital technologies into governance practices. At the same time, there was a thorough legal review of the Organic Act on Anti-Corruption B.E. 2561

Table 1. Key insights from literature on AI-driven public complaint processing.

| Topic | Main Points | References |
|---|---|---|
| Corruption and Governance Issues | Corruption undermines good governance, particularly in the developing world. In Thailand, the NACC is responsible for combating corruption but is hampered by inefficiency due to manual complaint processing. | [13–16] |
| Limitations of Current Complaint Handling | Manual processes result in lengthy average processing times, inconsistent decisions, and unequal workload. | [15–17] |
| AI Applications in Public Administration | AI is revolutionizing fields such as tax, healthcare, and legal. Such tools automate processes such as document classification and data extraction, increasing efficiency and transparency. | [18–22] |
| AI in Corruption Detection and Governance | AI has been employed in detecting corruption (e.g., procurement irregularities). Ethical issues need to be resolved when implementing AI in high-stakes governance applications. | [23–25] |
| Theoretical Framework: Public Value Theory | Focuses on the contribution of technology to strengthen legitimacy and democratic values, not only efficiency. Data strategies of Thailand are in place, but AI application in initial complaint handling is yet to be realized. | [16], [26] |
| Proposed AI-Based System Architecture | A modular system with: (1) OCR module for digitization of text; (2) NLP module for semantic processing and summarization; (3) ML classifier for multi-class prediction of types of complaints. | [27–29] |
| Research Questions | 1. Can AI cut down complaint screening time?<br>2. With what accuracy can AI classify corruption types?<br>3. What are the institutional implications? | |

carried out to ensure that any suggested AI intervention would be in line with current statutory requirements and data protection measures [30].

## Prototype system development

Phase II focused on the architectural design and iterative development of a modular AI-based prototype system with three interrelated modules. The first module, OCR, with customized preprocessing methods, was developed to translate scanned or handwritten complaint documents into machine-readable text [27]. It begins with the uploading of the image, which triggers a chain of preprocessing activities such as noise removal and grayscale conversion to clear the text. The system then performs line-by-line recognition, scanning the image in sequence and analyzing character patterns using Tesseract's Long Short-Term Memory (LSTM) neural network. The recognized text is decoded and formatted into structured output, like extracted text, confidence levels, and bounding box coordinates, and sent to downstream components for processing.

The OCR module, built on the Tesseract framework, was refined using a specialized Thai-language dataset comprising 2,892 printed documents containing over 14,000 characters across ten fonts. To ensure robustness and practical applicability, the model was also evaluated on 100 variably formatted handwritten and scanned samples, simulating real-world conditions commonly encountered in official complaint submissions.

The second module, an NLP Summarization Module, utilizes prompt-based inference with large language models (LLMs)—Gemini-1.5-Flash and Gemma27B, in this research—to extract salient factual content of Thai-language complaint texts and generate semantically meaningful summaries [28,29]. These pre-trained models, without additional training or fine-tuning, are guided using task-specific prompts to generate structured outputs across five essential dimensions: summary, accused entity, alleged misconduct, location, and monetary references. This normalized schema supports cloud and offline deployment models, respectively, to suit varying infrastructure and privacy needs. Auto-completion of relevant fields, indication of missing data, and single and batch submissions are handled by the system during processing. Structured outputs enable sorting, filtering, and visualization in real time, which works to disclose corruption patterns by actor, geography, or category. By automating the linguistic analysis, the module substantially reduces manual effort, improves consistency, and creates institutional capacity for evidence-based anti-corruption.

The third module, a classification module, uses Gemini-1.5-Flash and Gemma27B via prompt-based inference to categorize complaints into pre-defined categories of corruption-related misconduct. Rather than supervised machine learning or traditional model training, this approach draws upon meticulously designed prompts that ask the LLMs to perform classification tasks explicitly. This eliminates the need to train on a labeled dataset and allows the system to perform even in the absence of vast volumes of data. The use of general-purpose LLMs in this manner allows for rapid prototyping, reduces development complexity, and ensures flexibility in the face of evolving classification systems.

## System testing and performance assessment

The third stage focused on the empirical performance evaluation of the developed prototype within a controlled experimental environment. A total of 200 semi-synthetic complaint cases were initially generated to simulate the structure, content, and complexity of actual submissions received by the NACC. Following data cleaning and validation, 160 cases were retained for analysis, forming the final test dataset. The dataset comprised ten distinct complaint categories and contained no missing values, ensuring the reliability and completeness of the evaluation process. The overall distribution of the dataset across categories is presented in Table 2. Fig 1 shows the graphical representation of this distribution, while Fig 2 presents the data distribution in percentages for additional clarity.

The representative complaint cases were evaluated across three key dimensions: (1) Accuracy and related metrics including precision, recall, and F1-score, for text extraction and case classification; (2) System efficiency, assessed by

**Table 2. Distribution of the semi-synthetic data.**

| Rank | Category | Count | Count (%) |
|------|----------|-------|-----------|
| 1 | Abuse of Power | 54 | 33.75 |
| 2 | Procurement Fraud | 44 | 27.50 |
| 3 | Embezzlement | 16 | 10.00 |
| 4 | Unusual Wealth | 12 | 7.50 |
| 5 | Personnel Misconduct | 8 | 5.00 |
| 6 | Fraudulent Land Title | 7 | 4.38 |
| 7 | Bribery | 7 | 4.38 |
| 8 | Budget/Project Fraud | 7 | 4.38 |
| 9 | Conflict of Interest | 4 | 2.50 |
| 10 | Ethical Misconduct | 1 | 0.62 |

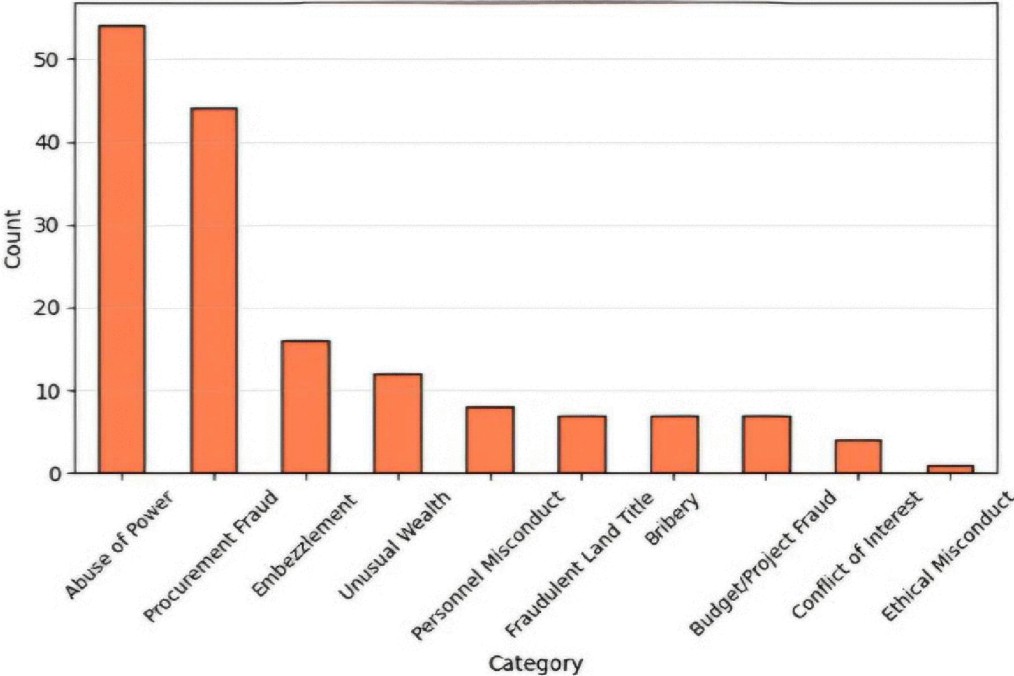

**Fig 1. Graphical representation of the data distribution.**

comparing automated processing times with manual review durations conducted by expert NACC personnel; and (3) User satisfaction, measured through structured post-evaluation surveys completed by 60 participants of respected operation departments. User perceptions were quantified using a five-point Likert scale based on utility, usability, and trust [29]. By integrating qualitative institutional insights with technical implementation, research ensured that the AI system was not only technically effective but also aligned with organizational processes, regulatory requirements, and end-user workflows. This socio-technical integration reflects a best-practice approach to AI deployment in the public sector, one that harmonizes algorithmic innovation with institutional norms, user competencies, and broader governing principles [31–33]. Fig 3 depicts the detailed analytical framework for this evaluation.

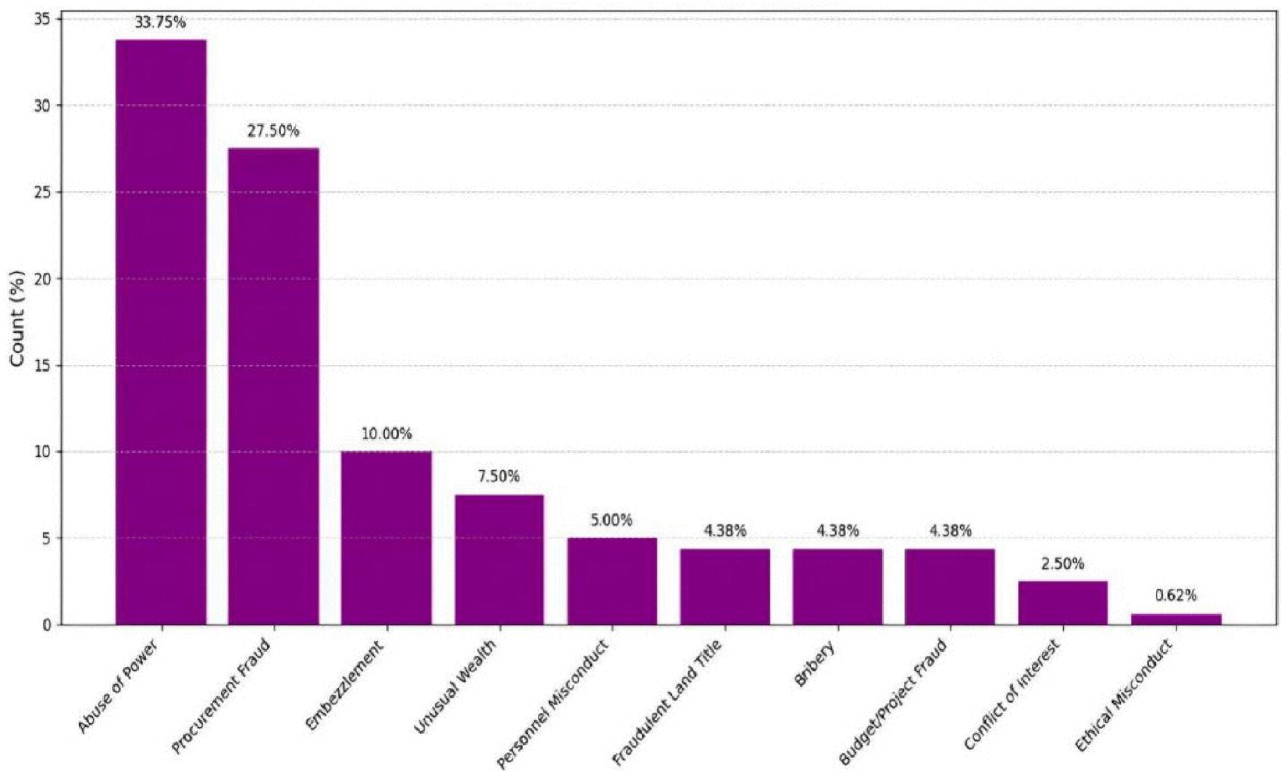

**Fig 2. Graphical representation of the data distribution (%).**

The study also identifies several challenges associated with linguistic and contextual complexity. Unstructured citizen reports, legally subtle expressions, and context-dependent ambiguities, particularly in multilingual settings, were found to reduce classification accuracy and precision. Additionally, the adaptation of general-purpose Large Language Models (LLMs) for Thai-language processing without fine-tuning, combined with the reliance on semi-synthetic data necessitated by the sensitivity of real-world cases, constrained the models' representativeness and overall performance.

For ethical consideration, it is important to note that all participants were properly informed about the objectives and procedures of the study before providing consent. Written consent was obtained through two methods: (i) an official announcement document shared with participants, and (ii) an on-screen agreement presented on the web application interface before the commencement of the test. In addition, verbal consent was obtained and witnessed during a pre-test meeting to further ensure participants' understanding and voluntary participation. No minors were involved in this study. A waiver from the ethics committee was unnecessary, as all participants provided informed consent.

## Analysis results

The AI prototype's performance was measured on three primary dimensions: accuracy, operational efficiency, and user satisfaction. To ensure realism and preserve the ethical expectations of data integrity, a customized test set of 160 semi-synthetic complaint cases was finally employed (as described in Table 2). The cases were created to reflect the structure and complexity of actual submissions to NACC.

The stand-alone, web-based OCR application was implemented using the Tesseract.js JavaScript library. Being capable of in-browser execution, it provides local uploading of images upon which the application client-side identifies

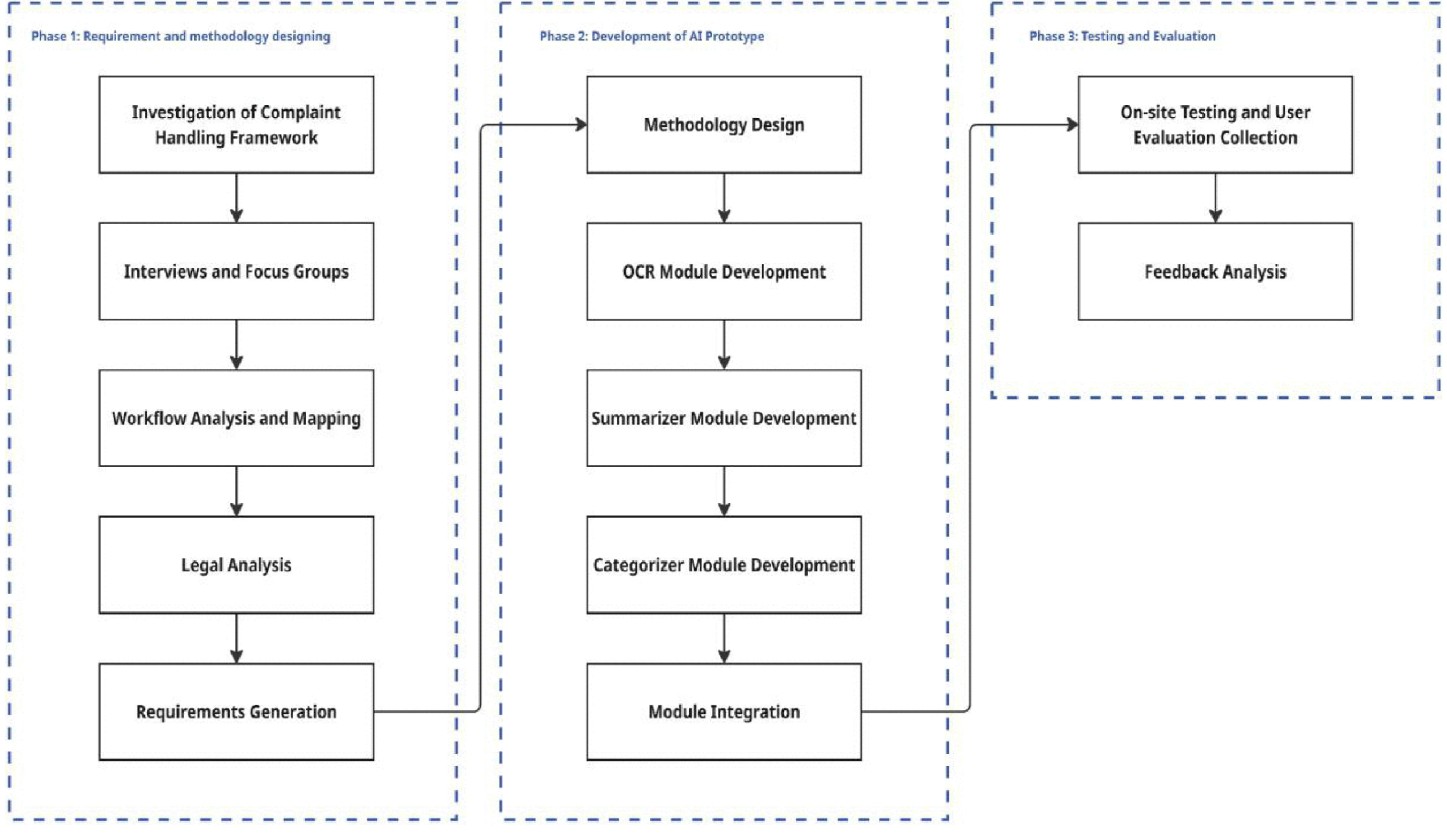

**Fig 3. Detailed analytical framework.**

and extracts text without recourse to server-side processing. The performance of the OCR module in this research was enhanced by training on a 2,892 printed document dataset with over 14,000 characters of text rendered in ten Thai fonts. Testing was performed on a representative test sample of 100 documents to simulate real-world variability in format and quality. The dataset comprised a balanced selection of handwritten text and scanned PDFs across a range of qualities to replicate the diversity and variety of real-world submissions. Fig 4 shows the steps involved in the OCR module, while Fig 5 presents an overview of the contents of the software. The model achieved an overall F1-score of 81.8%, precision of 84.2%, and a recall of 79.6%. For instance, for print text, the system achieved 72% word-level accuracy and 78% character-level accuracy. Detailed performance of the OCR module is exhibited in Table 3. As such, the findings provide an insight into the model's great potential in recognizing and extracting text from all types of formats, i.e., printed forms, scanned documents, and written notes, even those with moderate noise.

The model's performance is consistent with reports for other uses of Tesseract pipelines customized to public sector environments [34], which confirm its reliability to handle the variety of types of documents common to allegations submitted by citizens.

The NLP-based summarization module was thoroughly tested to assess its effectiveness in extracting and summarizing relevant information from unstructured legal documents. A panel comprising legal experts and NACC officials evaluated the clarity, completeness, and contextual accuracy of the generated summaries, assigning a mean score of 4.74 out of 5. The model consistently identified crucial details, such as the accused, actions undertaken, and relevant pieces of legislation. Notably, the automated summarization task achieved a 79% decrease in mean processing time compared to the

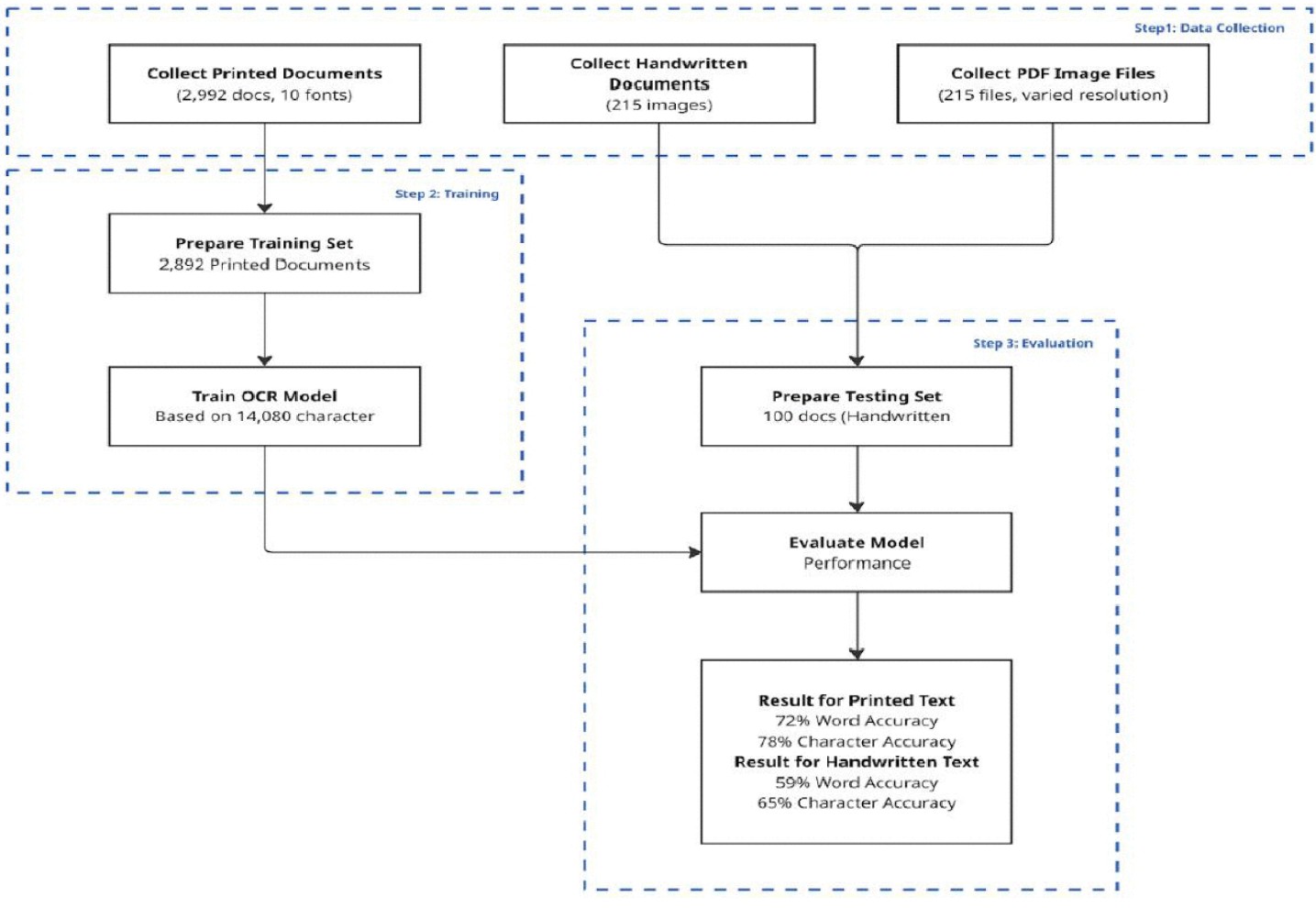

**Fig 4. OCR analysis flowchart.**

standard manual methods without compromising interpretability or accuracy. These results align with previous studies underscoring the significance of robust NLP design in policy and legal text analysis [21,33]. Fig 6 depicts the architecture and workflow of the module.

The classification module, as shown in Fig 7, achieved an overall accuracy of 57.50% in categorizing complaints into ten pre-specified legal categories (see Table 4). The model demonstrated particularly strong precision in identifying cases related to bribery and ethics violations (see Table 5). Beyond classification accuracy, the system significantly optimized operational efficiency, reducing the average processing time per complaint from 5.6 minutes under manual review to 71.47 seconds, representing a 78.6% decrease. This efficiency improvement highlights the potential of the system to significantly reduce human workload and alleviate case backlogs within the NACC's complaint-handling process. Fig 8 presents the confusion matrix used to derive performance metrics, accuracy, recall, and precision.

To enhance the reliability of the classification results, it was crucial to perform comparative benchmarking with recognized multilingual models. The performance of Gemini-1.5-Flash and Gemma-27B was compared with three established baselines: Zero-shot XLM-R, Zero-shot mDeBERTa, and Embedding Similarity (mMiniLM). These models embody distinct but complementary paradigms in multilingual natural language processing, cross-lingual zero-shot transfer (XLM-R),

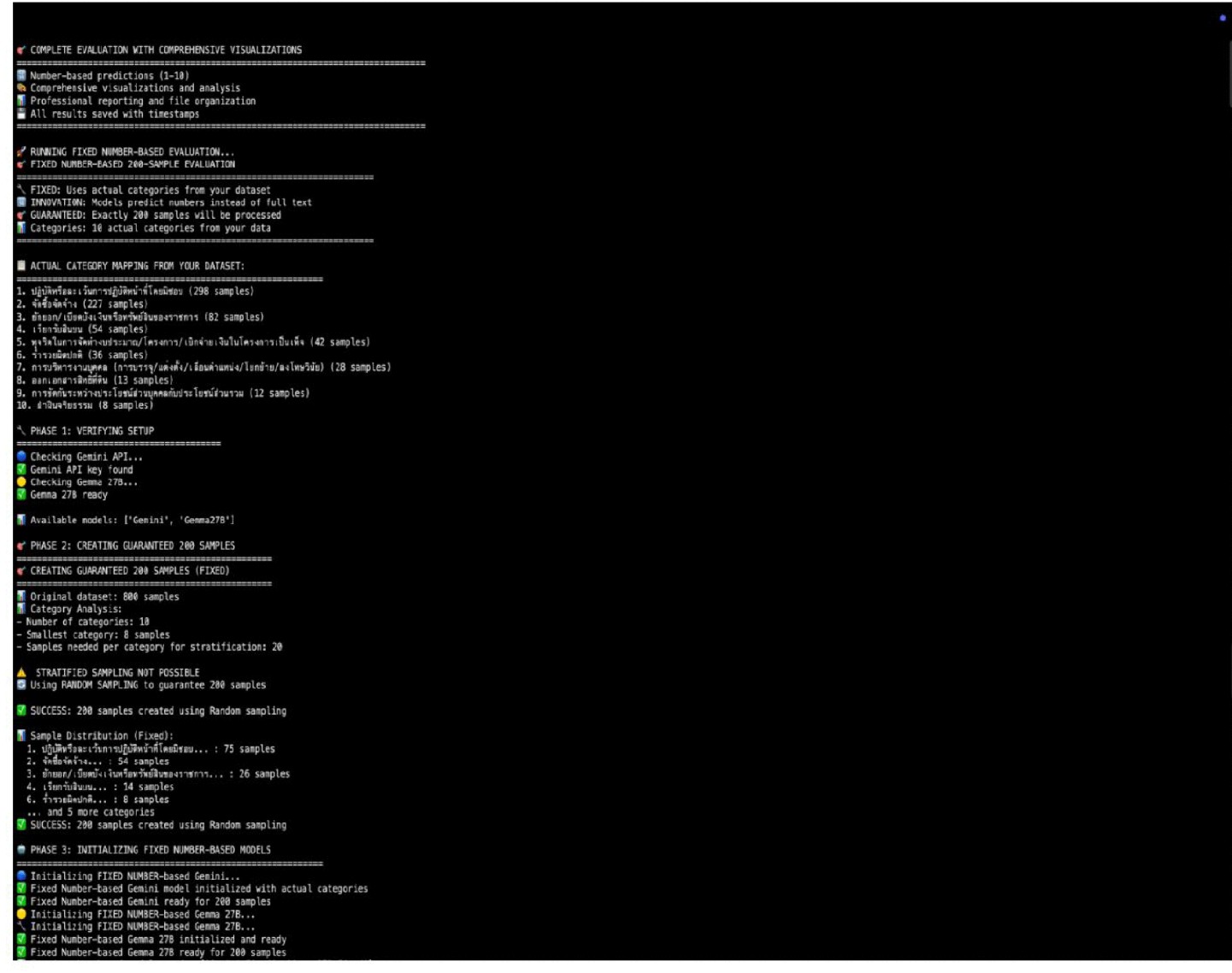

**Fig 5. A pictorial representation of the software contents.**

**Table 3. OCR module performance metrics across document types.**

| Document Type | Metric | Value (%) |
|---|---|---|
| Printed Text | Word Accuracy | 72 |
| | Character Accuracy | 78 |
| Handwritten Text | Word Accuracy | 59 |
| | Character Accuracy | 65 |
| Overall Result | F1-Score | 81.8 |
| | Precision | 84.2 |
| | Recall | 79.6 |

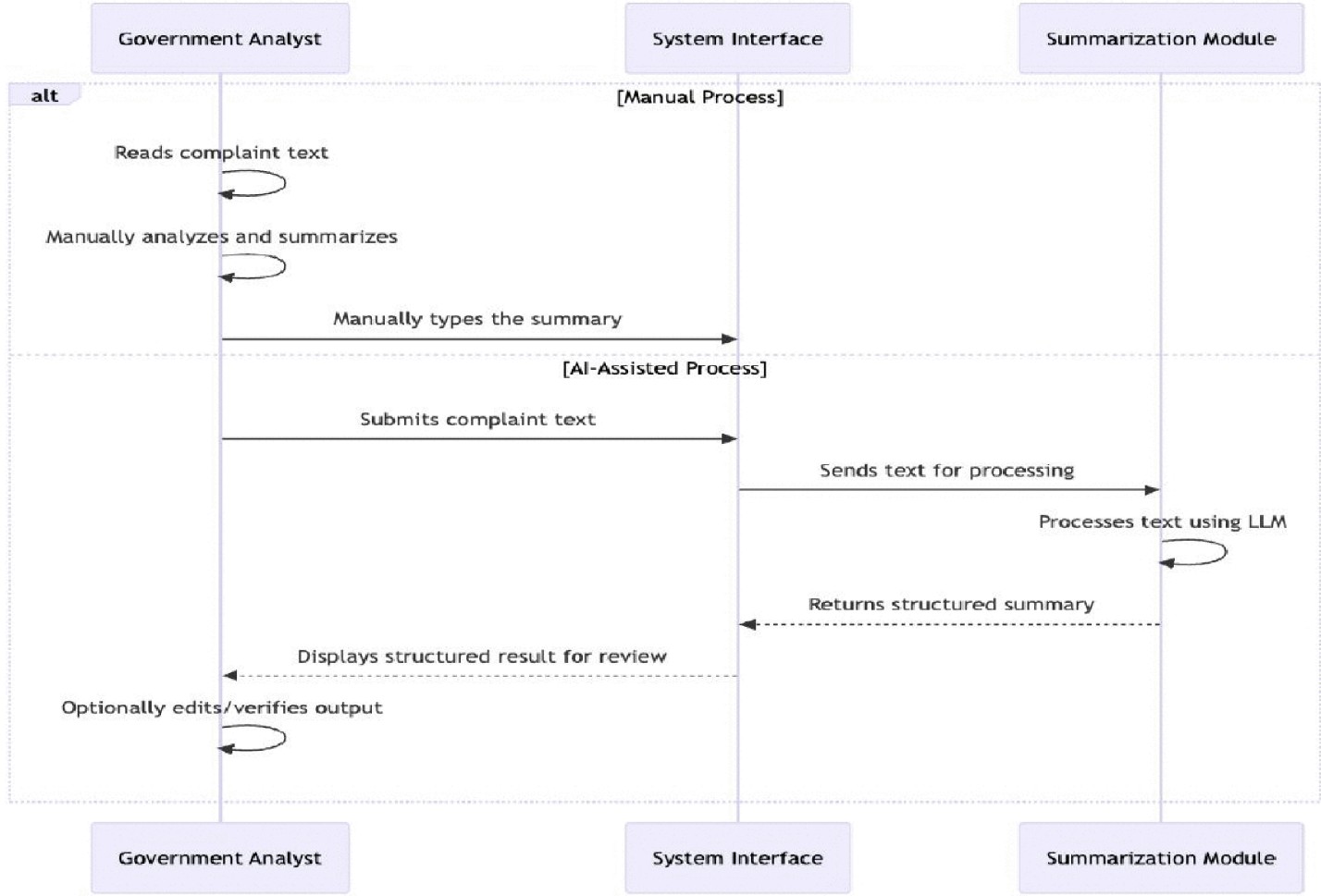

**Fig 6. Sequence diagram illustrating the workflow of the summarizer module.**

advanced encoder-based modeling (mDeBERTa), and lightweight semantic similarity representation (mMiniLM). Their inclusion offers reliable and rigorous benchmarks for evaluating large generative language models [35–37].

Both baseline and proposed models were evaluated on the same dataset of Thai corruption complaints. The baseline models, Zero-shot XLM-RoBERTa (XNLI), Zero-shot mDeBERTa-v3 (XNLI), and an embedding-based classifier with multilingual MiniLM, achieved modest performance, with macro-F1 scores spanning from 15% to 25% and accuracy falling between 16% and 23%. These findings indicate that standard multilingual transformers struggle to grasp the linguistic, cultural, and legal subtleties embedded in Thai complaint texts. In contrast, the proposed Gemini-1.5-Flash model attained a macro-F1 of 50.7% and an accuracy of 57.5%, outperforming the strongest baseline by more than double. The Gemma-27B model produced competitive results, achieving 45.0% macro-F1 and 52.0% accuracy (see Table 6). Collectively, these findings illustrate the superior capability of prompt-engineered large language models in handling domain-specific, multilingual legal texts without requiring fine-tuning or task-specific retraining.

The error analysis further revealed that the AI models excelled in classifying high-priority and linguistically distinct corruption types. For example, the Ethics Violation category achieved an ideal 100% F1 score, while categories like Bribery, Unusual Wealth, HR Mismanagement, and Embezzlement recorded F1 scores of 86.7%, 66.7%, and 62.7%, respectively,

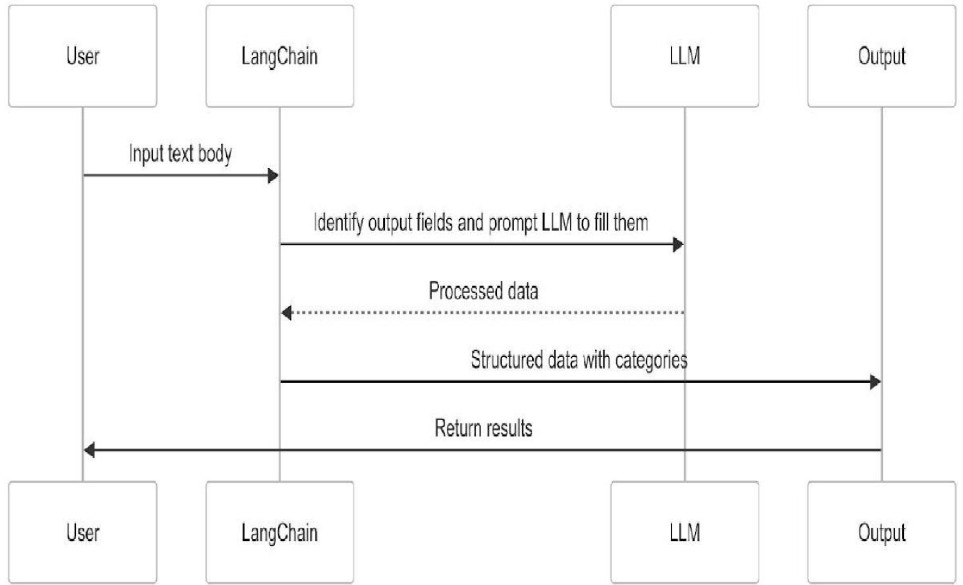

**Fig 7. Sequence diagram of the Classification module.**

**Table 4. Overall model performance of the classification module.**

| Metric | Gemini-1.5-flash (%) | Gemma27B |
|---|---|---|
| Accuracy | 57.5 | 52.0 |
| F1 Score (weighted) | 58.8 | 53.2 |
| F1 Score (Macro) | 50.7 | 45.0 |
| Precision | 52.5 | 46.8 |
| Recall | 53.1 | 48.1 |

with particularly high recall in Bribery cases. However, performance varied across categories because of class imbalance and the linguistic ambiguity inherent in complex administrative contexts. Precision was occasionally traded for recall in low-frequency categories; for instance, "Bribery" was frequently detected but sometimes misclassified, reflecting the trade-offs typical of contextual diversity in multilingual and legally nuanced text classification [38]. In summary, these results underscore the effectiveness of Gemini-1.5-Flash and Gemma-27B in enhancing the accuracy and interpretability of automated case screening in anti-corruption agencies, while also pointing out persistent challenges in addressing data imbalance and language ambiguity in public-sector AI applications

Efficiency in complaint handling is one of the most important measures in this study, quantified by comparative benchmarking against manual screening by trained NACC personnel. Fig 9 compares the processing time involved in manual screening and review with AI support, highlighting through their visual contrast their functional difference. As seen in Fig 10, the human process lasts about 333.39 seconds (~5.6 minutes) on average, while the AI-driven system performs the same task in 71.47 seconds (~1.2 minutes). This 78.6% reduction in processing time not only demonstrates the massive efficiency gains possible through automation but also reiterates the practicality of introducing AI to institutional screening processes in real-world applications.

Fig 11 shows the main performance metrics of the system—ease of use, precision, and speed—represented as average ratings and associated percentages. Based on an overall evaluation score of six, speed attained the highest rating at

**Table 5. Classification performance by category.**

| S/N | Category | Model | Precision | Recall | F1 Score | Sample |
|---|---|---|---|---|---|---|
| 1 | Misconduct in Duty | Gemini-1.5-flash | 0.607 | 0.493 | 0.544 | 75 |
| | | Gemma27B | 0.628 | 0.36 | 0.458 | 75 |
| 2 | Procurement Fraud | Gemini-1.5-flash | 0.633 | 0.574 | 0.602 | 54 |
| | | Gemma27B | 0.549 | 0.519 | 0.533 | 54 |
| 3 | Embezzlement | Gemini-1.5-flash | 0.64 | 0.615 | 0.627 | 26 |
| | | Gemma27B | 0.7 | 0.538 | 0.609 | 26 |
| 4 | Bribery | Gemini-1.5-flash | 0.813 | 0.929 | 0.867 | 14 |
| | | Gemma27B | 0.667 | 1 | 0.8 | 14 |
| 5 | Budget Fraud | Gemini-1.5-flash | 0.143 | 0.286 | 0.19 | 7 |
| | | Gemma27B | 0.118 | 0.571 | 0.195 | 7 |
| 6 | Unusual Wealth | Gemini-1.5-flash | 0.667 | 0.667 | 0.667 | 6 |
| | | Gemma27B | 0.444 | 0.667 | 0.533 | 6 |
| 7 | HR Mismanagement | Gemini-1.5-flash | 0.6 | 0.75 | 0.667 | 8 |
| | | Gemma27B | 0.5 | 0.75 | 0.6 | 8 |
| 8 | Illegal Land Documents | Gemini-1.5-flash | 0.5 | 0.75 | 0.6 | 4 |
| | | Gemma27B | 0.4 | 0.75 | 0.522 | 4 |
| 9 | Conflict of Interest | Gemini-1.5-flash | 0.333 | 0.333 | 0.333 | 3 |
| | | Gemma27B | 0.5 | 0.333 | 0.4 | 3 |
| 10 | Ethics Violation | Gemini-1.5-flash | 1 | 1 | 1 | 3 |
| | | Gemma27B | 0.75 | 1 | 0.857 | 3 |

87.67%, indicating a strong user agreement about the system's operational efficiency. User-friendliness followed closely at 84.83%, reflecting a generally positive user experience, while accuracy secured third place at 79%, showing strong results that still reveal chances for further refinement. The consistently high scores in every criterion collectively affirm the system's strong usability, reliability, and practical effectiveness in real-life applications.

From the point of view of the user, the system was overall highly rated by administrative staff. A 60-participant guided usability survey recorded a satisfaction mean score of 4.2 out of 5. The strengths in the respondents' feedback included how easy it was to use the interface, how logically the outputs were organized, and how well the system was consistent with institutional screening standards. Even with this general endorsement, some users advocated for the inclusion of additional premium features, most importantly, confidence-level tags to be inserted alongside classification results and a mechanism for human review in the event of borderline classification. These are echoed in literature on AI regulation, which demands human-in-the-loop systems as a means of enhancing transparency, accountability, and ethical resilience in automated decisions [39,40].

Overall, these findings reveal the technical feasibility and the institutional suitability of AI-enabled complaint screening at NACC. The system's modularity enables it to function either as a stand-alone triage system or as an add-on layer on existing infrastructures such as the Public E-Service for PESCA. PESCA, the online system of the NACC for receiving and processing corruption allegations by public servants and members of the public, already computerizes the reception of allegations but continues to be heavily dependent on human classification and assessment. The integration of AI-based modules, OCR, NLP, and auto-classification, would further boost the operational efficiency of PESCA through automation of preliminary screening and prioritization of cases of incoming allegations.

Such architectural adaptability not only supports scalability and interoperability between agencies but also aligns with Thailand's national e-governance and public sector modernization ambitions more broadly. More, however, is required, though, for successful and enduring deployment of such systems than thinking about technological maturity. Formal legal recognition of AI outputs, regulation facilitating regimes, and implementation of robust ethical protection measures are key

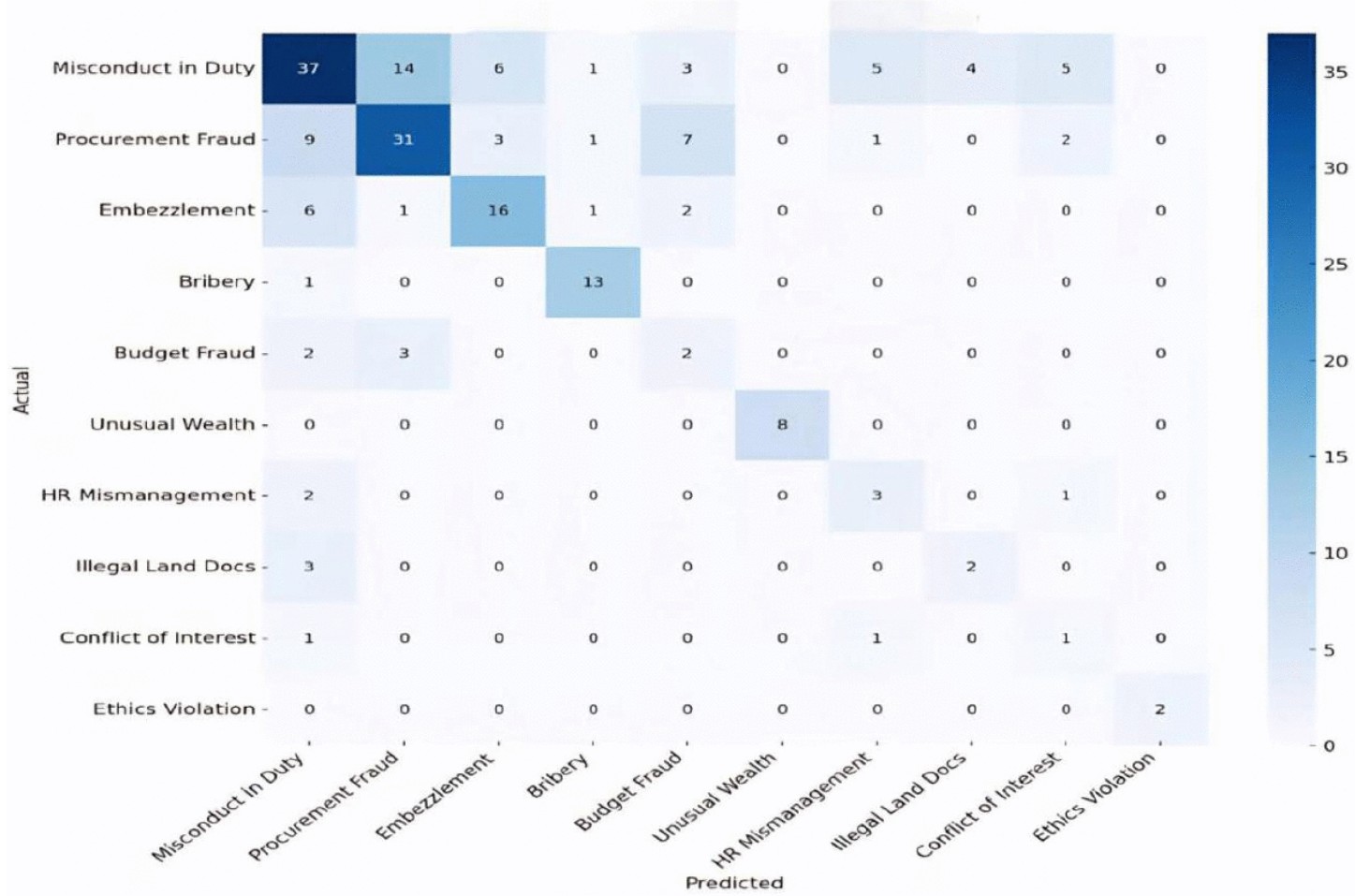

**Fig 8. The confusion matrix used to derive the performance metrics.**

**Table 6. Detailed comparative analysis of methods.**

| Model | Accuracy (%) | Precision (%) | Recall (%) | F1 (Macro %) | F1 (Weighted %) |
|---|---|---|---|---|---|
| Zero-shot XLM-R | 22.5 | 27.6 | 36.1 | 22.5 | 23.2 |
| Zero-shot mDeBERTa | 16.9 | 23.9 | 22.7 | 15.7 | 16.4 |
| Embedding Similarity (mMiniLM) | 23.1 | 23.4 | 43.9 | 25.1 | 19.6 |
| gemini-1.5-flash | 57.5 | 52.5 | 53.1 | 50.7 | 58.8 |
| Gemma27B | 52.0 | 46.8 | 48.1 | 45.0 | 53.2 |

enablers. These institutional designs are essential to procedural justice, public confidence, and ensuring adjudicative proceedings' legitimacy, especially in high-stakes corruption cases [41]. Meeting these foundational needs will be essential to moving toward the shift from prototype experiments to operational tools that substantively promote justice, transparency, and administrative integrity goals.

Transparency and explainability are constitutive imperatives in the deployment of artificial intelligence in government agencies, particularly in the sensitive domains of corruption determination. For public confidence and institutional

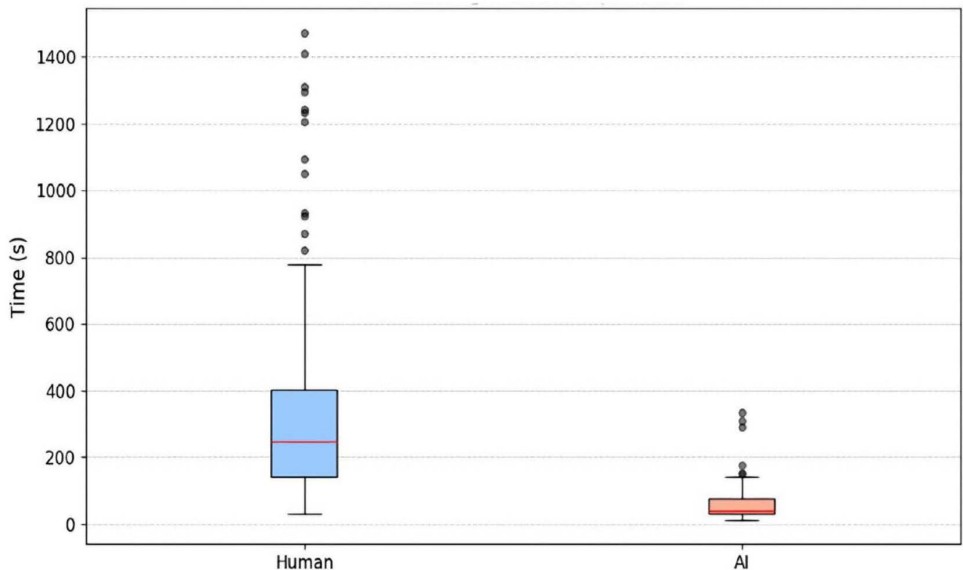

**Fig 9. Processing time comparison between human and AI processing.**

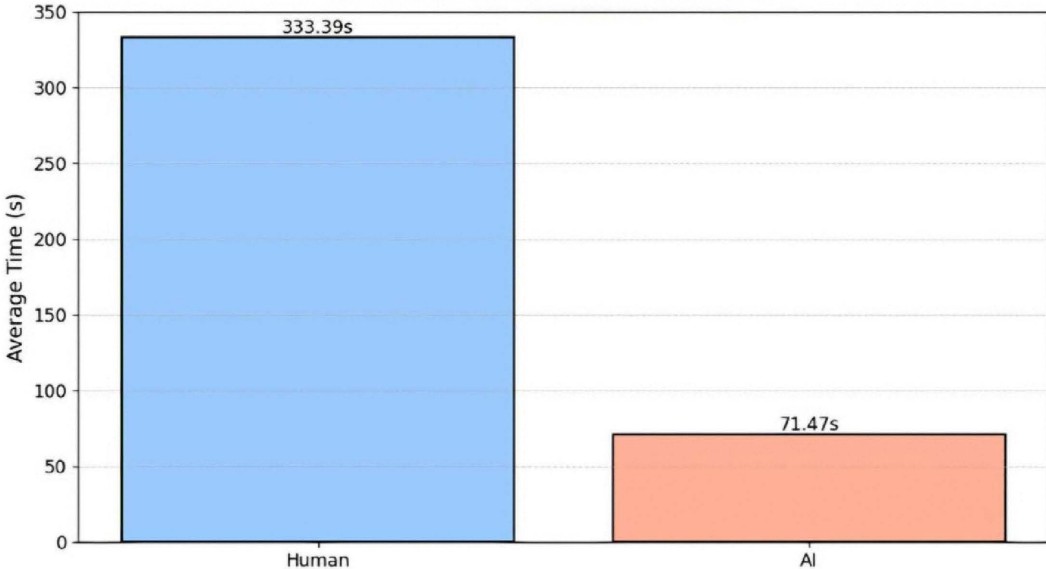

**Fig 10. Average processing time comparison.**

legitimacy, AI-aided allegation screening systems must entail explainable decision-making mechanisms that allow human investigators to inspect, authenticate, and—where necessary—overrule system suggestions. This interpretability demand is not technical but deeply normative in origin, rooted in democratic standards for accountability in automatic administrative decisions.

In addition, their outputs have to meet national anti-corruption legislation and be consistent with formal administrative procedures to be institutionally valid and admissible. Without such legislative conformity, the application of AI may produce

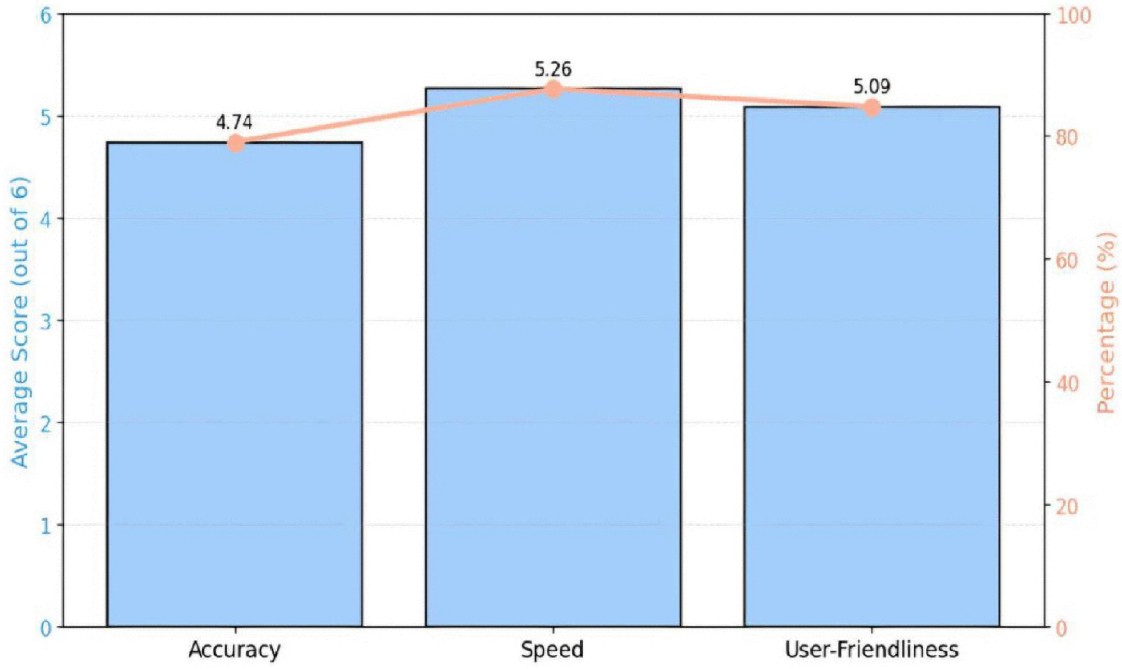

**Fig 11. System performance metrics.**

recommendations that are technically valid yet juridically unfirm or procedurally undecisive. To avoid these risks, formal regimes of regulatory oversight schemes must be instituted. Such frameworks would not only enshrine ethical and operational guidelines for AI application in anti-corruption efforts but also ensure redressal and review, and hence the robustness of due process.

There exists, however, an immediate institutional problem of the lack of a clear-cut legal framework for AI-driven decision-making in administrative settings. As there is a legal vacuum today, the products of AI systems can be interpreted as advisory, not authoritative, which restricts their real-world value and institutional adoption. As long as there are no codified instructions and statutory acceptance, the transformational value of AI for public administration will be hampered by procedural and legal uncertainty.

Equally important are longer-term organizational dynamics consequent on AI system adoption. Although initial user response has been consistently favorable, longer-term concerns of maintaining institutional trust in AI recommendations, redesigning staff tasks and workflows, and assigning legal responsibility for processes where they are automated require more research. These phenomena require firm, longitudinal research to determine what, in the longer term, institutionalizing AI does for organizational culture, decision-making structures, and public sector ethics.

Results show that AI-driven allegation screening yields remarkable improvements in terms of efficiency, including a reduction in processing time and better classification accuracy. In comparison to conventional systems such as PESCA, which rely heavily on human intuition in case triage, AI allows for evidence-based automated decision-making with no place for bias. Additionally, through the integration of machine learning systems capable of sorting and classifying complex or high-risk cases, the system allows for the deployment of resources strategically, making it possible for investigators to target complaints of greatest significance. All of these innovations collectively demonstrate not only that AI can be applied to improve the day-to-day working process but to achieve overall institutional goals, including greater transparency, procedural fairness, and public confidence in the integrity of the complaint-resolution process.

## Discussion and conclusion

This research confirms the technical feasibility and institutional applicability of deploying AI in the initial screening of corruption complaints in the government. By developing a modular AI prototype that combines OCR, NLP, and ML-based classification, the study justifies the digitization of complaint management procedures, improving accuracy, processing speed, and user satisfaction. The system's performance indicates that AI can meaningfully accelerate and standardize the intake and preliminary assessment of cases at the NACC, while reducing administrative burden.

Beyond technical performance, successful AI adoption requires organizational readiness, alignment with regulatory frameworks, and ethical governance, particularly in high-stakes domains such as anti-corruption, where automated decisions may carry legal and reputational consequences. Policy implementation must therefore include capacity building, stakeholder engagement, and adaptive regulations to ensure responsible and accountable use.

The modular design of the prototype also enhances its scalability and transferability. While tailored for the NACC, the system could be adapted to other Thai agencies with similar complaint-handling mandates, including the Office of the Ombudsman, the Public Sector Anti-Corruption Commission (PACC), and Local Administrative Organizations (LAOs). Internationally, the approach may benefit countries with similar bureaucratic systems, particularly in Southeast Asia and Latin America, where AI-driven complaint intake can enhance transparency, efficiency, and public trust when appropriately localized.

Despite these promising outcomes, several limitations merit attention. The system was tested on a semi-synthetic dataset, which may not fully capture the linguistic diversity, informal expressions, and subtle ambiguities present in real citizen submissions. Classification performance also varied across legal categories due to conceptually similar terms, indicating the need for enhanced domain-specific NLP and model enrichment. Moreover, deploying AI in public administration raises ethical and legal concerns, particularly around algorithmic bias, where historical imbalances in complaint data could influence prioritization. Addressing these challenges requires transparent decision-making, continuous monitoring, and rigorous auditing of datasets to safeguard fairness and accountability.

Going forward, future research should aim to enhance effectiveness while ensuring ethical, scalable, and contextually appropriate AI deployment. Key directions include: (i) localizing NLP components to incorporate legal and administrative terminology specific to target jurisdictions; (ii) ensuring interoperability with existing legacy systems to facilitate seamless integration; (iii) developing open-source, privacy-preserving implementations that comply with data protection laws; (iv) employing advanced deep learning and risk-scoring methods to improve classification performance and prioritize high-impact allegations; (v) establishing inter-agency data-sharing mechanisms to support coordinated investigations and end-to-end case tracking; and (vi) conducting long-term monitoring to evaluate the sustained impact on institutional processes, user confidence, adjudication outcomes, and public perception. By addressing these considerations, AI-driven systems can support efficient, transparent, and accountable governance while mitigating risks associated with bias, misclassification, and procedural fairness.

This research ultimately offers compelling proof that responsibly applied prompt-engineered large language models can greatly improve the screening of corruption complaints. The results underscore the potential of AI to enhance operational efficiency and bolster public trust, provided that ethical safeguards, policy alignment, and scalability considerations are systematically incorporated into implementation strategies.

## Acknowledgments

Sincere appreciation is extended to the National Anti-Corruption Commission (NACC) for their collaboration. The unwavering commitment and effort of the co-investigators were also instrumental in guiding the project from start to finish.

## Author contributions

**Conceptualization:** Issara Sereewatthanawut, Patipan Sriphon, Pattrawut Khunwipusit, Babatunde Oluwaseun Ajayi, Jutarat Suwaree, Wonlop Writthym Buachoom.

**Data curation:** Patipan Sriphon, Babatunde Oluwaseun Ajayi.

**Formal analysis:** Patipan Sriphon, Babatunde Oluwaseun Ajayi.

**Funding acquisition:** Issara Sereewatthanawut, Patipan Sriphon.

**Investigation:** Patipan Sriphon, Pattrawut Khunwipusit, Babatunde Oluwaseun Ajayi.

**Methodology:** Issara Sereewatthanawut, Patipan Sriphon, Pattrawut Khunwipusit, Babatunde Oluwaseun Ajayi.

**Project administration:** Issara Sereewatthanawut, Jutarat Suwaree, Wonlop Writthym Buachoom.

**Resources:** Issara Sereewatthanawut.

**Software:** Pattrawut Khunwipusit.

**Supervision:** Issara Sereewatthanawut, Babatunde Oluwaseun Ajayi, Jutarat Suwaree, Wonlop Writthym Buachoom.

**Validation:** Issara Sereewatthanawut, Pattrawut Khunwipusit, Ademola Enitan Ilesanmi.

**Visualization:** Patipan Sriphon, Pattrawut Khunwipusit, Ademola Enitan Ilesanmi.

**Writing – original draft:** Patipan Sriphon, Babatunde Oluwaseun Ajayi, Ademola Enitan Ilesanmi.

**Writing – review & editing:** Issara Sereewatthanawut.

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
