## [Decision Letter · Decision Letter 0]

5 Oct 2025

Dear Dr. Sereewatthanawut,

Thank you for submitting your manuscript to PLOS ONE. After careful consideration, we feel that it has merit but does not fully meet PLOS ONE’s publication criteria as it currently stands. Therefore, we invite you to submit a revised version of the manuscript that addresses the points raised during the review process.

We look forward to receiving your revised manuscript.

Kind regards,

Asif Khan, PhD Law

Academic Editor

PLOS ONE

Journal Requirements:

3. Please note that PLOS One has specific guidelines on code sharing for submissions in which author-generated code underpins the findings in the manuscript. In these cases, we expect all author-generated code to be made available without restrictions upon publication of the work. Please review our guidelines at https://journals.plos.org/plosone/s/materials-and-software-sharing#loc-sharing-code and ensure that your code is shared in a way that follows best practice and facilitates reproducibility and reuse.

National Research Council of Thailand

This research was supported by the National Research Council of Thailand (NRCT), whose support enabled the research to be carried out. We also thank the National Anti-Corruption Commission (NACC) at both the central and regional levels for their collaborative efforts, data provision, and constructive feedback, which confirmed that the system met real needs. Finally, we acknowledge the unfaltering commitment and effort of our co-investigators, whose dedication was crucial in guiding the project from start to finish.

National Research Council of Thailand

6. We note that your Data Availability Statement is currently as follows: All relevant data are within the manuscript and its Supporting Information files.

Reviewers' comments:

Reviewer's Responses to Questions

**Comments to the Author**

1. Is the manuscript technically sound, and do the data support the conclusions?

Reviewer #1: Yes

Reviewer #2: Yes

2. Has the statistical analysis been performed appropriately and rigorously?

Reviewer #1: Yes

Reviewer #2: Yes

3. Have the authors made all data underlying the findings in their manuscript fully available?

Reviewer #1: Yes

Reviewer #2: Yes

4. Is the manuscript presented in an intelligible fashion and written in standard English?

Reviewer #1: No

Reviewer #2: Yes

Reviewer #1: Dear Authors,

Your study makes a timely and valuable contribution by demonstrating how AI can improve the efficiency of corruption allegation screening at Thailand’s National Anti-Corruption Commission, with promising results on accuracy and significant reductions in processing time; however, to strengthen the manuscript, I suggest minor revisions: the abstract could better balance methods, results, and policy implications; the introduction should more clearly outline the research gap and draw on recent comparative studies on AI in governance; the methodology would benefit from additional detail on the dataset and language-specific challenges, while the results section could be enhanced with comparative benchmarks and a brief error analysis; finally, expanding the discussion to address policy implications, potential risks such as algorithmic bias, and scalability, along with polishing a few sentences for readability, will make the paper even stronger and more impactful.

Reviewer #2: The publication entitled "Efficient AI-Driven Allegation Screening: A Case Study of Thailand’s National Anti-Corruption Commission" offers a pertinent and insightful examination of the integration of artificial intelligence into anti-corruption processes. The case study methodology is well implemented, providing both pragmatic insights and scholarly significance. The emphasis on Thailand’s National Anti-Corruption Commission provides a significant regional viewpoint that is frequently overlooked in AI governance research.

The study exhibits methodological rigor, good organization, and a robust understanding of AI systems and institutional settings. The discourse on screening efficiency, data prioritizing, and algorithmic openness is exceptionally well-expressed. Furthermore, the study examines critical ethical and operational issues without exaggerating the potential of AI, so fostering a fair and realistic evaluation.

The manuscript is complete and requires no modifications. The writing is lucid and professional, the figures and tables are helpful, and the references are current and relevant.

This is a significant addition to the domains of AI in governance and anti-corruption strategies. I endorse the text for acceptance in its current form, requiring only little copyediting for language consistency.

**Do you want your identity to be public for this peer review?** For information about this choice, including consent withdrawal, please see our Privacy Policy

Reviewer #1: **Yes:** Sidra kanwel, Assistant professor, Sarhad institute of legal studies, Sarhad university of science and IT Pakistan

Reviewer #2: **Yes:** MASEEH ULLAH

---

## [Author Response · Author response to Decision Letter 1]

12 Nov 2025

Editor

Comment:

Please ensure that your manuscript meets PLOS ONE's style requirements, including those for file naming

Response:

The manuscript has been revised in accordance with the PLOS ONE’s style guidelines.

Comment:

Please provide additional details regarding participant consent. In the ethics statement in the Methods and online submission information, please ensure that you have specified (1) whether consent was informed and (2) what type you obtained (for instance, written or verbal, and if verbal, how it was documented and witnessed). If your study included minors, state whether you obtained consent from parents or guardians. If the need for consent was waived by the ethics committee, please include this information.

Response

All participants were properly informed about the objectives and procedures of the study before providing consent. Written consent was obtained through two methods: (i) an official announcement document shared with participants, and (ii) an on-screen agreement presented on the web application interface before the commencement of the test. In addition, verbal consent was obtained and witnessed during a pre-test meeting to further ensure participants' understanding and voluntary participation. No minors were involved in this study. A waiver from the ethics committee was unnecessary, as all participants provided informed consent. This statement is provided in the methodology section (see page 17) and also has been included in the online submission system.

Comment:

Please note that PLOS One has specific guidelines on code sharing for submissions in which author-generated code underpins the findings in the manuscript. In these cases, we expect all author-generated code to be made available without restrictions upon publication of the work. Please review our guidelines at https://journals.plos.org/plosone/s/materials-and-software-sharing#loc-sharing-code and ensure that your code is shared in a way that follows best practice and facilitates reproducibility and reuse.

Response:

The codes and related sources for this project can be accessed at https://github.com/VAP-Solution/kpi_accusation_fieldtest, which contains materials related to the field test. The test employed a custom web application developed using Appsmith, available at https://www.vapsolution.app/nacc_llm_fieldtest, which served as the primary interface for testing and data collection. However, the research data cannot be made publicly available due to data protection agreements with the data provider, the National Anti-Corruption Commission (NACC) of Thailand.

Comment:

Thank you for stating the following financial disclosure:

National Research Council of Thailand

Response:

Comment:

Thank you for stating the following in the Acknowledgments Section of your manuscript:

This research was supported by the National Research Council of Thailand (NRCT), whose support enabled the research to be carried out. We also thank the National Anti-Corruption Commission (NACC) at both the central and regional levels for their collaborative efforts, data provision, and constructive feedback, which confirmed that the system met real needs. Finally, we acknowledge the unfaltering commitment and effort of our co-investigators, whose dedication was crucial in guiding the project from start to finish.

National Research Council of Thailand

Response:

All funding-related text has been removed from the manuscript, and the acknowledgement section has been revised accordingly

Funding:

This research received funding from the National Research Council of Thailand (NRCT), Thailand

Comment:

We note that your Data Availability Statement is currently as follows: All relevant data are within the manuscript and its Supporting Information files.

Response:

The codes and related sources for this project can be accessed at https://github.com/VAP-Solution/kpi_accusation_fieldtest, which contains materials related to the field test. The test employed a custom web application developed using Appsmith, available at https://www.vapsolution.app/nacc_llm_fieldtest, which served as the primary interface for testing and data collection. However, the research data cannot be made publicly available due to data protection agreements with the data provider, the National Anti-Corruption Commission (NACC) of Thailand.

Comment:

Response:

During the revision process, additional relevant studies were cited to strengthen the content of the paper, particularly in the Introduction and Results sections. The newly added references are highlighted in red for ease of identification.

Reviewer 1

Comment:

Your study makes a timely and valuable contribution by demonstrating how AI can improve the efficiency of corruption allegation screening at Thailand’s National Anti-Corruption Commission, with promising results on accuracy and significant reductions in processing time; however, to strengthen the manuscript, I suggest minor revisions: the abstract could better balance methods, results, and policy implications; the introduction should more clearly outline the research gap and draw on recent comparative studies on AI in governance; the methodology would benefit from additional detail on the dataset and language-specific challenges, while the results section could be enhanced with comparative benchmarks and a brief error analysis; finally, expanding the discussion to address policy implications, potential risks such as algorithmic bias, and scalability, along with polishing a few sentences for readability, will make the paper even stronger and more impactful

Response:

The abstract has been revised to provide a balanced presentation of the methods, results, and policy implications (see pages 1-2). The introduction has been thoroughly updated to clearly articulate the research gap and incorporate comparative studies on the use of AI in governance (see pages 3-5). The methodology section has been carefully expanded to include additional details on the dataset and language-specific challenges (see pages 12-17). The results section now presents a more detailed comparison of findings, while the explanation of the error analysis has been refined for greater clarity (see pages 24-26). Furthermore, the discussion section has been comprehensively revised to address policy implications, potential risks, and scalability (see pages 32-34). Overall, the entire manuscript has undergone extensive revision to improve clarity, coherence, and conciseness.

Reviewer 2

Comment:

The publication entitled "Efficient AI-Driven Allegation Screening: A Case Study of Thailand’s National Anti-Corruption Commission" offers a pertinent and insightful examination of the integration of artificial intelligence into anti-corruption processes. The case study methodology is well implemented, providing both pragmatic insights and scholarly significance. The emphasis on Thailand’s National Anti-Corruption Commission provides a significant regional viewpoint that is frequently overlooked in AI governance research.

The study exhibits methodological rigor, good organization, and a robust understanding of AI systems and institutional settings. The discourse on screening efficiency, data prioritizing, and algorithmic openness is exceptionally well-expressed. Furthermore, the study examines critical ethical and operational issues without exaggerating the potential of AI, so fostering a fair and realistic evaluation.

The manuscript is complete and requires no modifications. The writing is lucid and professional, the figures and tables are helpful, and the references are current and relevant.

This is a significant addition to the domains of AI in governance and anti-corruption strategies. I endorse the text for acceptance in its current form, requiring only little copyediting for language consistency.

Response:

The entire manuscript has been carefully revised and thoroughly proofread to ensure consistency and clarity.

---

## [Editor Report · Decision Letter 1]

25 Nov 2025

Efficient AI-Driven Allegation Screening: A Case Study of Thailand’s National Anti-Corruption Commission

PONE-D-25-44792R1

Dear Dr. Sereewatthanawut,

We’re pleased to inform you that your manuscript has been judged scientifically suitable for publication and will be formally accepted for publication once it meets all outstanding technical requirements.

Kind regards,

Asif Khan, PhD Law

Academic Editor

PLOS ONE
---

## [Editor Report · Acceptance letter]

PONE-D-25-44792R1

PLOS One

Dear Dr. Sereewatthanawut,

I'm pleased to inform you that your manuscript has been deemed suitable for publication in PLOS One. Congratulations! Your manuscript is now being handed over to our production team.

Kind regards,

on behalf of

Dr. Asif Khan

Academic Editor

PLOS One